# WT1-AS/IGF2BP2 Axis Is a Potential Diagnostic and Prognostic Biomarker for Lung Adenocarcinoma According to ceRNA Network Comprehensive Analysis Combined with Experiments

**DOI:** 10.3390/cells11010025

**Published:** 2021-12-23

**Authors:** Mingxi Jia, Yi Shi, Yang Xie, Wen Li, Jing Deng, Da Fu, Jie Bai, Yushui Ma, Zavuga Zuberi, Juan Li, Zheng Li

**Affiliations:** 1Hunan Key Laboratory of Processed Food for Special Medical Purpose, College of Food Science and Engineering, Central South University of Forestry and Technology, Changsha 410004, China; 20200100079@csuft.edu.cn (M.J.); 20210100081@csuft.edu.cn (Y.S.); 20213553@csuft.edu.cn (Y.X.); fuda@tongji.edu.cn (D.F.); T20071464@csuft.edu.cn (J.B.); T20031458@csuft.edu.cn (J.L.); 2College of Life Sciences and Chemistry, Hunan University of Technology, Zhuzhou 412007, China; 3Central Laboratory for Medical Research, Shanghai Tenth People’s Hospital, Tongji University School of Medicine, Shanghai 200072, China; mayushui@tongji.edu.cn; 4Department of Science and Laboratory Technology, Dar es Salaam Institute of Technology, Dares Salaam P.O. Box 2958, Tanzania; Tanzania.zuberiz@nm-aist.ac.tz; 5NHC Key Laboratory of Carcinogenesis, Cancer Research Institute and School of Basic Medical, Central South University, Changsha 410013, China; lizheng@csu.edu.cn

**Keywords:** WT1-AS/IGF2BP2 axis, ceRNA network, novel biomarkers, prognosis, lung adenocarcinoma

## Abstract

Lung adenocarcinoma (LUAD) is one of the most common malignancies, and there is still a lack of effective biomarkers for early detection and prognostic prediction. Here, we comprehensively analyze the characteristics of. an RNA sequencing data set of LUAD samples. In total, 395 long non-coding RNAs (lncRNAs), 89 microRNAs (miRNAs), and 872 mRNAs associated with c-Myc were identified, which were differentially expressed between tumor and normal tissues. The most relevant pathway was found to be WT1-AS–miR-200a-3p–IGF2BP2 according to the rules of competitive endogenous RNA (ceRNA) regulation. WT1-AS and IGF2BP2 expression were positively correlated and increased in LUAD samples, while miR-200a-3p had relatively low expression. The high expression of WT1-AS and IGF2BP2 was associated with poor prognosis in LUAD patients, while low expression of miR-200a-3p predicted reduced survival (*p* < 0.05). The analysis of the multi-gene regulation model indicated that the WT1-AS (downregulation)–miR-200a-3p (upregulation)–IGF2BP2 (downregulation) pattern significantly improved the survival of LUAD patients. Finally, reverse transcription-polymerase chain reaction (RT-PCR) and Western blotting were detected in LUAD cells, and the results are consistent with the bioinformatics analysis. In summary, the WT1-AS/IGF2BP2 axis is a potential prognostic biomarker in LUAD and is expected to become an effective target for diagnosis and treatment.

## 1. Introduction

Lung cancer is a malignant tumor with the highest morbidity and mortality in the world, and it is a serious threat to human health [1,2]. Lung adenocarcinoma (LUAD) is the most common type of lung cancer, with more than 1 million deaths from LUAD worldwide each year [3,4,5]. Therefore, effective prognostic biomarkers and therapeutic targets for LUAD must be explored to improve our deficiencies in disease diagnosis, prevention, and treatment [6].

Long non-coding RNA (lncRNA) and microRNA (miRNA) are the most studied non-coding RNAs, and they play an important role in tumor diagnosis and treatment. LncRNAs play an important role in regulating gene expression by binding to chromatin regulatory proteins, and they can interact with other RNAs and proteins to alter chromatin modification and transcriptional or post-transcriptional gene regulation [7,8,9]. In recent years, the competitive endogenous RNAs (ceRNAs) network has revealed a novel mechanism of RNA interactions [10,11]. In molecular biology, lncRNA can act as ceRNA and regulate other RNA transcripts by competing for shared miRNAs. In detail, lncRNA competes directly or indirectly with miRNA for binding, which will eventually lead to a weakened interaction between miRNA and mRNA [12]. It has been proven to be involved in the development and progression of lung [13], pancreatic [14], liver [15], and ovarian cancers [16]. For example, SNHG5 regulates the expression of SOX4 by binding miR-489-3p, thereby regulating the proliferation and apoptosis of acute myeloid leukemia [17]. BBOX1-AS1 upregulates the expression of SH2B1 by sponging miR-361-3p, thereby promoting the development of colorectal cancer [18]. However, the role of the lncRNA-related ceRNA regulatory network in LUAD remains unclear, and the establishment of a ceRNA network is very important for the prognosis prediction and treatment decision-making of LUAD patients [19,20].

The protein product of the proto-oncogene c-Myc is a transcription factor that regulates a wide range of cellular processes [21]. c-Myc genes play key roles in the regulation of many biological processes, such as the cell cycle, proliferation, apoptosis, protein synthesis, and cell metabolism [22]. Analysis of 26 cancer types (3131 cancer samples) indicated that amplification of the human c-Myc gene occurred in most types of malignancies, highlighting the key role of c-Myc in tumor pathogenesis [23]. The latest research indicates that in addition to a large number of protein-coding genes, many lncRNA and miRNA are also downstream targets of c-Myc [24]. It has been reported that c-Myc is overexpressed in several tumor types and targets the lncRNA DANCR, which in turn promotes cancer cell proliferation [25]. LncRNA CCAT1 was determined to regulate c-Myc mRNA expression by sponging miR-155 [26]. At present, the specific regulation mechanism of the c-Myc-related ceRNA network in LUAD has not yet been fully clarified.

In this study, we aimed to construct a ceRNA network associated with c-Myc in LUAD. We conducted a systematic and comprehensive study on the RNA-seq data of LUAD from The Cancer Genome Atlas (TCGA). The differentially expressed lncRNAs, miRNAs, and mRNAs related to c-Myc were screened and a ceRNA network was constructed. Then, a triple regulatory network of WT1-AS–miR-200a-3p–IGF2BP2 was identified by expression analysis and survival analysis. The diagnostic and prognostic value of IGF2BP2 was obtained by univariate analysis in LUAD. The multi-gene interaction regulation analysis model was used to group WT1-AS–miR-200a-3p–IGF2BP2 (upregulation to “+”, downregulation to “−”) into eight different combinations of expression patterns and to analyze the correlation between different expression patterns and OS of LUAD patients. Finally, the expression of WT1-AS, IGF2BP2, and miR-200a-3p in lung adenocarcinoma cell lines was verified by RT-PCR. Therefore, our study may help to understand the underlying molecular mechanisms of LUAD and provide new targets for the treatment of LUAD.

## 2. Materials and Methods

### 2.1. Data Preparation and Processing

The RNA profiles and clinical information of human LUAD samples were downloaded from the TCGA database (https://portal.gdc.cancer.gov/, accessed on 18 January 2021). The lncRNA, miRNA, and mRNA information were extracted from them separately. By matching mRNA and miRNA samples, 515 tumor samples and 57 cancer samples were finally determined. The miRNA codes were converted to mature miRNA names using the Starbase v2.0 database (http://starbase.sysu.edu.cn, accessed on 20 January 2021).

### 2.2. Screening of Differentially Expressed Genes

The 515 LUAD patients were divided into a c-Myc^low^ group (c-Myc expression ≤ 2541.5, *n* = 258) and c-Myc^high^ group (c-Myc expression > 2541.5, *n* = 257) according to the median expression of c-Myc (c-Myc expression value = 2541.5). Then, the lncRNA, miRNA, and mRNA data were also divided into a c-Myc^low^ group and c-Myc^high^ group according to the serial number of the LUAD patients. The Perl and R languages were used to analyze and process the RNA sequencing data. Differentially expressed genes were analyzed in c-Myc^high^ and c-Myc^low^ LUAD samples through the R package “edgeR”, and the “Trimmed Mean of M-values” (TMM) normalization method was used to calibrate the read length of the genes. The differentially expressed lncRNAs had thresholds of |log_2_FC| > 0.8 and *p* < 0.01, the differentially expressed miRNAs had thresholds of |log_2_FC| > 0.3 and *p* < 0.01, and the differentially expressed mRNAs had thresholds of |log_2_FC| > 0.5 and *p* < 0.01. While performing differential expression analysis between LUAD samples and adjacent nontumorous samples, the critical value of differentially expressed lncRNAs was set to |log_2_FC| > 0.8 and *p* < 0.01, the critical value of differentially expressed miRNAs was set to |log_2_FC| > 0.3 and *p* < 0.01, and the critical value of differentially expressed mRNAs was set to |log_2_FC| > 0.8 and *p* < 0.01. Only RNAs that were jointly verified as differentially expressed in both analysis methods were identified as candidate genes for subsequent analysis.

### 2.3. Construction of the ceRNA Network in LUAD

The ceRNA network was constructed according to the following steps: (1) The mircode database (http://www.mircode.org/, accessed on 22 January 2021) was used to predict potential miRNAs for lncRNAs targeting. (2) Two databases, miRDB (http://www.mirdb.org/miRDB/, accessed on 22 January 2021) and Targetscan database (http://www.targetscan.org/, accessed on 22 January 2021), were used to identify the downstream targets (mRNAs) of miRNAs to improve the accuracy of the target gene prediction results. Only mRNAs that were simultaneously identified in the two databases were confirmed as candidate genes for subsequent analysis. (3) The target genes were compared with the differentially expressed mRNAs obtained from the previous analysis using the VennDiagram package in R software and the overlapping genes were selected for the next step of analysis, resulting in miRNA–mRNA interaction pairs. (4) We constructed the lnRNA-miRNA-mRNA ceRNA network by integrating lncRNA–miRNA pairs with miRNA–mRNA pairs and visualized by Cytoscape v3.7.0.

### 2.4. Survival Analysis and Constructing a Specific Prognostic Model of LUAD

The lncRNAs, miRNAs, and mRNAs in the ceRNA network were analyzed by Kaplan–Meier and log-rank analysis to determine their relationship with overall survival (OS) in LUAD patients. The relationship was analyzed by univariate and multivariate Cox regression analysis between candidate genes and OS, which was used to determine the biological prognosis markers. The WT1-AS–miR-200a-3p–IGF2BP2 ceRNA regulation axis was finally obtained, and *p* < 0.05 was deemed significant. In addition, the correlation between different expression combinations patterns of three genes (WT1-AS, miR-200a-3p, IGF2BP2) and OS with LUAD patients was analyzed. WT1-AS, miR-200a-3p, and IGF2BP2 were divided into high expression groups (“+”) and low expression groups (“−”) according to the median of each gene expression value. Kaplan–Meier analysis and the log-rank test were performed to analyze the effect of different expression patterns of the WT1-AS–miR-200a-3p–IGF2BP2 axis on OS in LUAD patients.

### 2.5. Function Abundance Analysis

To understand the possible biological processes and pathways of the ceRNA network, functional enrichment of coding genes was performed by Gene Ontology (GO) and Kyoto Encyclopedia of Genes and Genomes (KEGG). Then, the top 100 genes associating with the IGF2BP2 gene in LUAD were obtained from GEPIA (http://gepia.cancer-pku.cn/, accessed on 28 January 2021). These genes were pathway analyzed by DAVID (https://david.ncifcrf.gov/tools.jsp, accessed on 28 January 2021) including biological process (BPs), the cellular component (CC), and molecular function (MF), and visualized by “ggplot2”.

### 2.6. RNA Extraction and Quantitative

Three lung adenocarcinoma cell lines (A549, PC-9 and H1299) and one normal bronchial epithelial cell line (BEAS-2B) were sponsored by the Xiangya Hospital Central South University (Changsha, China). A549, PC-9, and H1299 cells were cultured in RPMI medium 1640 and BEAS-2B cells were cultured in Dulbecco’s Modified Eagle’s medium (Thermo Fisher Scientific, Waltham, MA, USA) supplemented with 10% fetal bovine serum (Thermo Fisher Scientific, Waltham, MA, USA) and 1% penicillin-streptomycin (Sangon Biotech, Shanghai, China). All cells were cultured and kept in a humid incubator under standard conditions. Total RNA was extracted utilizing the FastPure Cell/Tissue Total RNA Isolation Kit (Vazyme, Nanjing, China) according to the manufacturer’s instructions. The resulting RNA (2µg) was used as a template for reverse-transcribing to first-strand cDNA synthesis of mRNA and lncRNA by using a BeyoRT™II First Strand cDNA Synthesis Kit (Beyotime, Shanghai, China) in a total volume of 20 µL. Real-time quantitative polymerase chain reaction (RT-qPCR) was performed by the CFX96 Real Time PCR system (Bio-rad, Hercules, CA, USA) using a BeyoFast™ SYBR Green qPCR Mix kit (Beyotime, Shanghai, China) referring to the protocol. The cDNA of miRNA was generated using a miRNA 1st Strand cDNA Synthesis Kit (Vazyme, Nanjing, China) and stem-loop primers. Glyceraldehyde-3-phosphate dehydrogenase (GAPDH) was used as the internal control, and. mRNA and lncRNA values were normalized to that of GAPDH. U6 was used as an internal control for miRNA. The relative expression of RNA was calculated by the 2^−ΔΔCt^ method. Each sample was repeated in triplicate. The primer sequences were as follows:GAPDH, F: 5′-CAGGAGGCATTGCTGATGAT-3′,R: 5′-GAAGGCTGGGGCTCATTT-3′.WT1-AS, F: 5′-CGTTTGGAGGACCGAGCATCAG-3′,R: 5′-TGGCATAATTGGACCGCACAGTAG-3′.miR-200a-3p, F: 5′-GCGCGTAACACTGTCTGGTAA-3′,R: 5′-AGTGCAGGGTCCGAGGTATT-3′,IGF2BP2, F: 5′-GATGAACAAGCTTTACATCGGG-3′,R: 5′-GATTTTCCCATGCAATTCCACT-3′.

### 2.7. Protein Extraction and Western Blotting Analysis

Cells were lysed on ice with radioimmunoprecipitation assay (RIPA) lysis buffer (Aolarbio Life Sciences, Beijing, China) containing protease inhibitors. Then, they were centrifuged at high speed and the supernatant was collected. An equal volume of 2× sodium dodecyl sulfate polyacrylamide gel electrophoresis (SDS-PAGE) loading buffer was added to the supernatant liquid, mixed well, and heated at 95 °C. The specific experimental details of Western blotting refer to previous research methods [27]. The GAPDH protein was used as an internal reference. Polyclonal antibody GAPDH (#20035) was purchased from Forevertek Biotechnology CO., Ltd. (Changsha, China). Polyclonal antibody IGF2BP2 was purchased from Thermo Fisher Scientific (Waltham, MA, USA).

### 2.8. Immune Infiltrate Levels and Expression Analysis of IGF2BP2

We applied TIMER (https://cistrome.shinyapps.io/timer/, accessed on 10 February 2021) to research the association between tumor-infiltrating immune cells and expression of IGF2BP2 in LUAD. We explored the correlation between IGF2BP2 expression levels and the abundance of tumor-infiltrating immune cells as well as the prognostic value of LUAD. In addition, we analyzed the correlation between IGF2BP2 and 16 tumor-infiltrating immune cell markers.

### 2.9. Statistical Analysis

GraphPad Prism (version 8.0) and SPSS 23.0 software (SPSS Inc., Chicago, IL, USA) were used for data statistics and processing in this study. The Mann–Whitney test and independent *t*-test were used to analyze the differences between the two groups of data, and one-way ANOVA analysis was used to compare multiple groups of data. *p* < 0.05 was considered statistically different.

## 3. Results

### 3.1. High Expression and Prognostic Value of c-Myc in LUAD

We first analyzed the possible role of c-Myc in LUAD. Based on The Human Protein Atlas (HPA) database, we found that c-Myc is over-expressed in lung cancer tissues (Figure 1A and Appendix A). The OncoPrint graph of the cBioPortal database shows that the changes in c-Myc gene expression in the TCGA LUAD dataset are mainly due to its amplification (Figure 1B). The immunohistochemical (IHC) staining from THPA also confirmed a similar c-Myc expression imbalance (Figure 1C), and the patient data are listed in Appendix A. We analyzed and investigated the clinical significance of high c-Myc expression in LUAD patients. The result showed that the c-Myc expression level of LUAD was positively related to the TNM stage (*p* = 0.0297) and tumor size (*p* = 0.0228) of the patients (Figure 1D,E).

### 3.2. Identification of Differentially Expressed Genes

To establish the lncRNA-miRNA-mRNA triple regulatory network, we firstly identified the differentially expressed genes in LUAD samples with c-Myc^high^ and c-Myc^low^ expression groups as well as in LUAD and adjacent-normal tissues. In total, 641 differentially expressed lncRNAs (341 upregulated and 299 downregulated), 105 differentially expressed miRNAs (36 upregulated and 69 downregulated), and 1259 differentially expressed mRNAs (730 upregulated and 529 downregulated) were sorted out in LUAD samples with c-Myc^high^ and c-Myc^low^ expression groups. While a total of 4289 differentially expressed lncRNAs (3343 upregulated and 945 downregulated), 409 differentially expressed miRNAs (303 upregulated and 105 downregulated), and 6737 differentially expressed mRNAs (4485 upregulated and 2252 downregulated) were identified between LUAD samples and normal lung tissue samples. The distribution of differentially expressed lncRNAs, miRNAs, and mRNAs were visualized by the volcano plot as shown in Figure 2A. A total of 395 lncRNAs, 89 miRNAs, and 872 mRNAs were jointly identified in the c-Myc^high^ and c-Myc^low^ expression groups as well as in LUAD and adjacent-normal tissue groups.

### 3.3. Construction of the ceRNA Regulatory Network

To establish the lncRNA-miRNA-mRNA ceRNA network in LUAD, we performed combined analysis in the c-Myc^high^ and c-Myc^low^ expression groups as well as in LUAD and adjacent-normal tissue groups. Potential miRNAs targeting lncRNAs were identified through the tarbase database, and finally 34 lncRNAs and 8 miRNAs with targeting effects were identified. Two databases, miRDB and TargetScan, were used to identify the downstream targets (mRNA) of these eight miRNAs to improve the accuracy of target gene prediction results. Only mRNAs that were simultaneously identified in the two databases were confirmed as candidate genes for subsequent analysis. The results revealed that 52 out of 872 differentially expressed mRNAs were identified. Finally, the lncRNA-miRNA-mRNA ceRNA network associated with c-Myc was constructed using Cytoscape software, including 34 lncRNAs, 8 miRNAs, and 52 mRNAs (Figure 2B). In order to further explore the potential functions of the ceRNA network related to c-Myc, functional enrichment analysis was performed of these mRNAs by using Metascape (https://metascape.org/gp/index.html#/main/step1, accessed on 28 January 2021). The results displayed that mRNAs that participate in the network were specially enriched in the “trans-synaptic signaling”, “regulation of system process”, and “flavonoid glucuronidation” (Figure 2C).

### 3.4. Selection of ceRNA Prognostic Models with LUAD Specificity

To identify ceRNA networks of significant prognostic value in LUAD, we first performed OS analysis of LUAD patients using Kaplan–Meier analysis and the log-rank test. Totally, four lncRNAs (LINC00460, AP002478.1, RMRP, WT1-AS), three miRNAs (miR-200a-3p, miR-206, miR-508), and eight mRNAs (PAQR9, C11orf86, IGF2BP2, NOS1, GALNT13, PI15, RFX4, RIC3) were found to be associated with OS based on *p* < 0.05 (Figure 3 and Appendix A). Furthermore, we analyzed the expression levels of 15 RNAs in LUAD samples with c-Myc^high^ and c-Myc^low^ groups as well as in LUAD and adjacent-normal lung tissues. Our results demonstrated that one downregulated (RMRP) and three upregulated (LINC00460, MUC20-OT1, WT1-AS) lncRNAs, three upregulated (miR-200a-3p, miR-206, miR-508) miRNAs, two downregulated (PI15, RIC3), and six upregulated (PAQR9, C11orf86, IGF2BP2, NOS1, GALNT13, RFX4) mRNAs in LUAD samples with c-Myc^high^ and c-Myc^low^ expression groups (Figure 3A and Appendix A). While four upregulated (LINC00460, MUC20-OT1, RMRP, WT1-AS) lncRNAs, two upregulated (miR-200a-3p, miR-508) and one downregulated (miR-206) miRNAs, and four upregulated (PAQR9, C11orf86, IGF2BP2, PI15) and four downregulated (NOS1, GALNT13, RFX4, RIC3) mRNAs in LUAD and adjacent-normal lung tissues (Figure 3B and Appendix A). In addition, the expression levels of these RNAs were verified in 57 (or 46) paired LUAD samples from the TCGA cohort (Appendix A).

Finally, the lncRNA (WT1-AS, upregulated)–miRNA (miR-200a-3p, downregulated)–mRNA (IGF2BP2 and PAQR9, upregulated) regulatory network associated with the overall survival of LUAD patients was determined. The above analysis indicated that lncRNA WT1-AS may act as ceRNA to enhance the expression of IGF2BP2 and PAQR9 through sponge miR-200a-3p (Figure 4A). Based on pairing between miR-200a-3p and the target site in the IGF2BP2, PAQR9 and WT1-AS were predicted by TargetScan and MiRcode, respectively (Figure 4B). Through expression correlation analysis, we discovered that WT1-AS expression was positively correlated with IGF2BP2 expression, and miR-200a-3p expression was negatively correlated with IGF2BP2 expression. At the same time, the correlation analysis results also showed a significant positive relationship between c-Myc expression and WT1-AS and IGF2BP2 expression, while a negative relationship with miR-200a-3p expression (Figure 4C). However, the expression of PAQR9 has a poor correlation with the expression of WT1-AS and miR-200a-3p. Therefore, the WT1-AS–miR-200a-3p–IGF2BP2 regulatory axis in the ceRNA network may be an important potential prognostic model and for further analysis.

### 3.5. Effect of Different Expression Patterns of the WT1-AS–miR-200a-3p–IGF2BP2 Axis on OS of LUAD

In addition, the correlation between different expression combinations patterns of the WT1-AS–miR-200a-3p–IGF2BP2 axis and overall survival (OS) with LUAD patients was analyzed. The WT1-AS, miR-200a-3p, and IGF2BP2 were divided into high expression groups (“+”) and low expression groups (“−”) according to the median of each gene expression value. Eight combined expression patterns were ultimately obtained, including: W^+^/m^−^/I^+^, W^−^/m^+^/I^−^, W^−^/m^−^/I^−^, W^+^/m^+^/I^+^, W^+^/m^−^/I^−^, W^+^/m^+^/I^−^, W^+^/m^+^/I^−^, W^−^/m^−^/I^+^, W^−^/m^+^/I^+^, and W^−^/m^+^/I^+^. These results showed that the W^−^/m^+^/I^−^ pattern significantly improved the OS of patients compared to the W^+^/m^−^/I^+^ expression pattern (*p* = 0.0007) (Figure 4D,E). In addition, other expression patterns indicated that the W^+^/m^+^/I^+^, W^+^/m^−^/I^−^, and W^−^/m^+^/I^+^ pattern improved the OS of LUAD patients relative to the W^+^/m^−^/I^+^ pattern (W^+^/m^+^/I^+^ vs. W^+^/m^−^/I^+^, *p* = 0.0144; W^+^/m^−^/I^−^ vs. W^+^/m^−^/I^+^, *p* = 0.03; W^−^/m^+^/I^+^ vs. W^+^/m^−^/I^+^, *p* = 0.0231). These findings suggested that inhibition of WT1-AS and IGF2BP2 expression levels or upregulation of miR-200a-3p expression may both halt cancer cell progression and improve the prognosis of LUAD patients. These data are consistent with the analysis results in Figure 4, which proves the accuracy of the multi-gene comprehensive analysis results.

### 3.6. Clinical Significance of the WT1-AS–miR-200a-3p–IGF2BP2 Axis in LUAD Patients

In order to known whether the expression levels of WT1-AS–miR-200a-3p–IGF2BP2 axis was affected by clinical characteristics, the correlation was explored between WT1-AS, miR-200a-3p, and IGF2BP2 expression with clinical factors. These results demonstrated that the expression of both WT1-AS and IGF2BP2 was positively associated with tumor size (WT1-AS: *p* = 0.0449, IGF2BP2: *p* = 0.0434), and IGF2BP2 expression were also positively correlated with TNM stage (*p* = 0.0464). The expression of miR-200a-3p was negatively related to lymph-node metastasis (*p* = 0.0474) and TNM stage (*p* = 0.0026). We further analyzed the correlation between clinical features and OS. Some traditional prognostic factors of LUAD patients in the TCGA cohort are closely related to OS, including TNM stage, tumor size, lymph-node metastasis, and distant metastasis (*p* < 0.01, Appendix A). Furthermore, we analyzed the correlation between these clinical features and the survival of LUAD patients to further reveal their prognostic significance in LUAD. In the multivariate regression analysis model, TNM stage (*p* = 0.043), tumor size (*p* = 0.034), and lymph node metastasis (*p* < 0.0001) were still related to the OS of LUAD patients (Appendix A). In summary, WT1-AS/IGF2BP2 may serve as an important prognostic factor for patients with LUAD.

In order to further explore the possible function of IGF2BP2 in LUAD, Gene Ontology (GO) and Kyoto Encyclopedia of Genes and Genomes (KEGG) enrichment analysis were performed on the top 100 IGF2BP2-related genes in LUAD (Figure 5). The enrichment term associated with IGF2BP2 was “RNA transport”. In addition, GO enrichment analysis associated with IGF2BP2 was mainly enriched in “cell division”, “protein binding”, and “nucleoplasm”.

### 3.7. Validation of Regulatory Axis Gene Expression

To better comprehend the mechanism of the WT1-AS–miR-200a-3p–IGF2BP2 axis in LUAD, the expression of these genes in lung adenocarcinoma cell lines was tested in vitro. The RT-PCR results showed that both WT1-AS and IGF2BP2 were significantly upregulated in H1299, PC-9, and A549 cells relative to BEAS-2B cells, and the highest expression was found in PC-9 cells (Figure 6). The expression of miR-200a-3p was significantly downregulated in H1299 and A549 relative to normal lung cells. At the same time, IGF2BP2 protein expression was also upregulated in lung adenocarcinoma cell lines relative to normal lung cells. These results were consistent with the conclusions of the bioinformatics analysis.

### 3.8. Correlation between Immune Infiltration and Expression of IGF2BP2 in LUAD

We used TIMER analysis to assess the potential relationship between IGF2BP2 expression and the level of immune infiltration in LUAD. “SCNA” module analysis indicated that several immune cell infiltration levels are related to the change of the IGF2BP2 gene copy number, including B cells, CD4+ T cells, neutrophils, and dendritic cells, in LUAD (Appendix A). The “Gene” module analysis demonstrated that IGF2BP2 expression seemed to be positively correlated with the level of infiltration of neutrophils and CD4+ T cells in LUAD (Appendix A). In addition, we also explored the association between IGF2BP2 and 16 immune cell markers. The results demonstrated that the expression levels of 6 markers (KIR2DL4, STAT1, GATA3, STAT5A, PD-1, GZMB) from natural killer cell and Th1, Th2, and T cell exhaustion were significantly positively correlated with the expression of IGF2BP2 in LUAD (Appendix A). These results indicate that the WT1-AS/IGF2BP2 axis may be related to the level of tumor-infiltrating immune cells in LUAD. Furthermore, we evaluated the association between the immune infiltration level and the clinical prognosis of LUAD patients. Interestingly, low levels of B cells (*p* < 0.0001), CD8+ T cells (*p* = 0.049), neutrophils (*p* = 0.043), and dendritic cells (*p* = 0.017) were associated with poor prognosis of LUAD patients (Appendix A).

## 4. Discussion

The ceRNA regulatory network has been revealed to be involved in the development and progression of many human cancers, including lung cancer [28], breast cancer [29], and liver cancer [15]. The c-Myc gene has been shown to be a key oncogene affecting many malignancies including lung adenocarcinoma [30,31,32]. Therefore, our study aimed to establish a triple network of c-Myc-associated ceRNAs with prognostic relevance in LUAD, based on the fact that c-Myc was recognized as a key cancer gene. Studies have shown that c-Myc could be involved in the development of NSCLC as a direct target gene of miR-449c, where overexpression of miR-449c could inhibit lung tumor growth in vivo [33]. EGFR promotes lung tumorigenesis by activating c-Myc, which in turn stimulates the expression of miR-7 [34]. Studies have indicated that compared with normal lung epithelial cells BEAS-2B, the c-Myc gene is highly expressed in lung adenocarcinoma cells A549 and H1299 [35,36]. Our analysis indicated that the highly malignant stage of lung adenocarcinoma corresponds to higher c-Myc expression levels, and high c-Myc expression was related to larger tumor size.

The lncRNA (WT1-AS)–miRNA (mir-200a-3p)–mRNA (IGF2BP2) regulatory axis associated with LUAD prognosis was identified based on the survival-related analysis and differential expression results of RNAs. Although previous studies have extensively reported the role of WT1-AS in different types of cancer, the role of WT1-AS in tumors remains incompletely defined [37,38]. For example, downregulation of WT1-AS can promote the proliferation and invasiveness of gastric cancer cells [39]. WT1-AS is downregulated in non-small-cell lung cancer (NSCLC) tissues, and it predicted the poor survival rate of NSCLC patients [40]. However, these findings do not seem to apply to LUAD and are somewhat different from our findings. This study found that WT1-AS was highly expressed in LUAD tissues and related to poor prognosis of LUAD. MiR-200a-3p is the most dysregulated member of the miR-200 family, which is closely associated with several types of cancer [41,42]. Overexpression of miR-200a induces downregulation of hepatocyte growth factor (HGF), which in turn reduces NSCLC cell migration and promotes apoptosis [43]. In addition, miR-200a-3p was downregulated in NSCLC tissues and cell lines [44]. The expression level of miR-200a-3p was related to tumor size, tumor stage, and lymph-node metastasis in NSCLC. All these results are consistent with the findings of our analysis. Curiously, we also discovered that miR-200a-3p expression was significantly higher in LUAD tissues than in paired non-tumor tissues, contrary to other assays and analyses, and we consider that this may be related to the smaller paired sample size (*n* = 46). The prediction of the matching site between miR-200a-3p and the target gene confirms that it was the hub of the WT1-AS–miR-200a-3p–IGF2BP2 regulatory axis. The gene encoding IGF2BP2 has been found to be amplified and overexpressed in many human malignancies. For example, overexpression of IGF2BP2 enhances proliferation, invasion, and metastasis of glioblastoma multiforme cells [45]. Moreover, miR-485-5p overexpression led to a decrease in IGF2BP2, which in turn inhibited the growth and metastasis of mouse A549 cells [46]. Our findings also demonstrated that high expression of IGF2BP2 was related to poor prognosis in LUAD, and the expression of IGF2BP2 was positively correlated with the TNM stage and tumor size. In addition, IGF2BP2 is positively correlated with CD4+ T cells and neutrophil levels, which contradicts the results of the correlation analysis between immune infiltration and OS. The relationship between IGF2BP2 and tumor immune infiltration remains to be determined.

The RT-PCR results confirmed that both WT1-AS and IGF2BP2 were significantly upregulated in lung adenocarcinoma cell lines relative to the normal bronchial epithelial cell line BEAS-2B, and the highest expression was found in PC-9 cells. The expression of miR-200a-3p was significantly downregulated in H1299 and A549 relative to normal lung cells. At the same time, IGF2BP2 protein expression was also upregulated in lung adenocarcinoma cell lines relative to normal lung cells. These results were consistent with the conclusions of the bioinformatics analysis. We also analyzed the correlation between different expression patterns of the WT1-AS–miR-200a-3p–IGF2BP2 axis and LUAD patients with OS. The results of the survival analysis showed that the W^−^/m^+^/I^−^ pattern significantly improved the OS compared to the W^+^/m^−^/I^+^ expression pattern in LUAD patients (*p* = 0.0007). The other expression pattern also confirmed that downregulation of WT1-AS and IGF2BP2 or upregulation of miR-200a-3p obviously affected the prognosis of LUAD patients. Combining the analysis results of this study, we suggest that lncRNA WT1-AS may compete with miR-200a for binding, which will eventually lead to a weakened interaction between miR-200a and IGF2BP2 and thus promote the LUAD process.

Of course, there are inevitably still some limitations to our research. First of all, our study data were obtained from the TCGA database and there may be potential biases in the analysis due to missing clinicopathological parameters. Secondly, this study lacks experimental validation on the mechanisms of ceRNA regulation. Therefore, more clinical studies are needed to confirm our hypothesis further. Despite the limitations of this study, we creatively combined the key tumor gene-Myc with the ceRNA network for comprehensive analysis. We determined that the ceRNA-based WT1-AS/IGF2BP2 pathway may be a potential therapeutic target and prognostic biomarker of LUAD. In short, this research further explored the mechanism of LUAD from the molecular level, and discovered new diagnostic and prognostic biomarkers, which have guiding significance for the treatment and continued research of LUAD.

## 5. Conclusions

In this study, we conducted a detailed comprehensive analysis of the c-Myc-related ceRNA network in LUAD. We determined that the high expression of WT1-AS and IGF2BP2 is related to the poor prognosis of patients with LUAD, while the low expression of miR-200a-3p predicts a lower survival rate. The expressions of WT1-AS, IGF2BP2, and miR-200a-3p were verified in the LUAD cell line by RT-PCR and Western blotting, which were consistent with the results of the bioinformatics analysis. In conclusion, the WT1-AS/IGF2BP2 pathway may be a potential therapeutic target and prognostic biomarker of LUAD. We will conduct more clinical studies in follow-up studies to further confirm our hypothesis.

## Figures and Tables

**Figure 1 cells-11-00025-f001:**
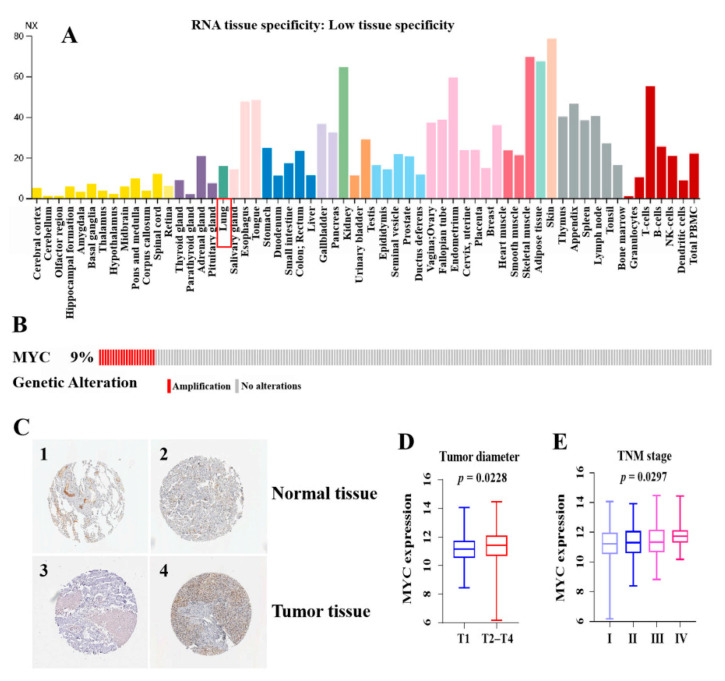
High expression of c-Myc in LUAD and prognostic value. (**A**) c-Myc expression distribution in all tissues. (**B**) The distribution of c-Myc genomic alterations in TCGA LUAD is shown on cBioPortal OncoPrint plots. (**C**) Validation of c-Myc expression at the translational level by the HPA database, normal tissue (1 and 2), and tumor tissue (3 and 4). (**D**) Correlation analysis of c-Myc expression values with tumor size (T1 ≤ 3 cm, T2–T4 > 3 cm). (**E**) Correlation analysis of c-Myc expression values with TNM stage. The RNAs expression value was logarithmized with log2. *p* < 0.05 was deemed as statistically significant.

**Figure 2 cells-11-00025-f002:**
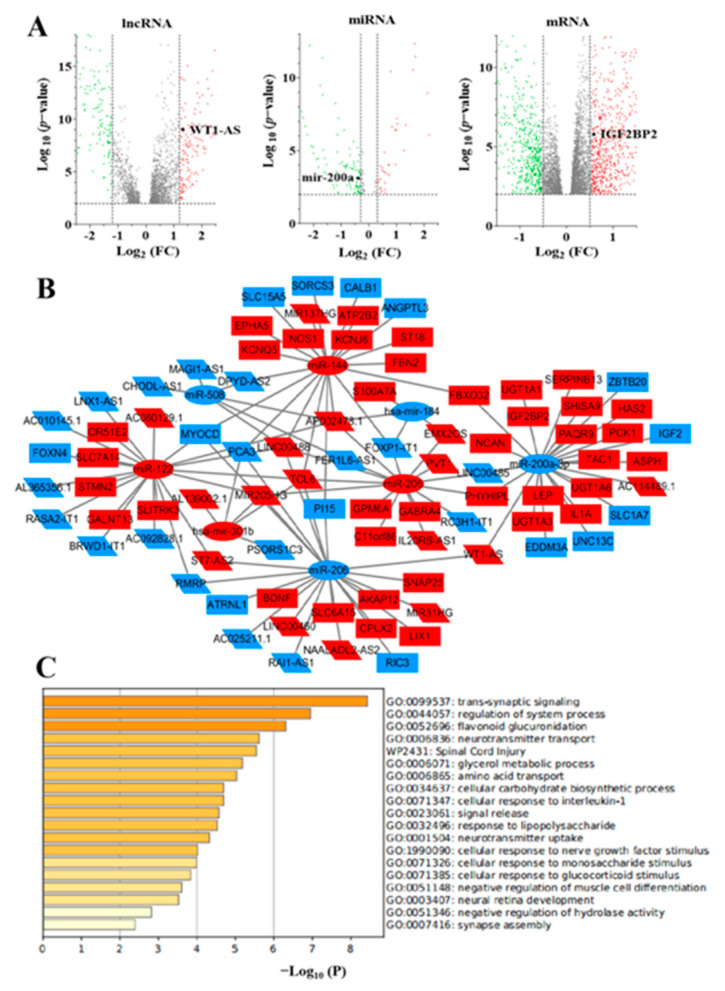
The c-Myc-related ceRNA network in LUAD. (**A**) The vocano plot of lncRNAs, miRNAs and mRNAs. Red stands for upregulation and green stands for downregulation. (**B**) The c-Myc-related ceRNA regulatory network in LUAD. The oval represents miRNAs, the diamond represents lncRNAs, and the rectangle represents mRNAs. Red stands for upregulation and blue stands for downregulation. (**C**) Functional enrichment analysis of mRNAs in the network.

**Figure 3 cells-11-00025-f003:**
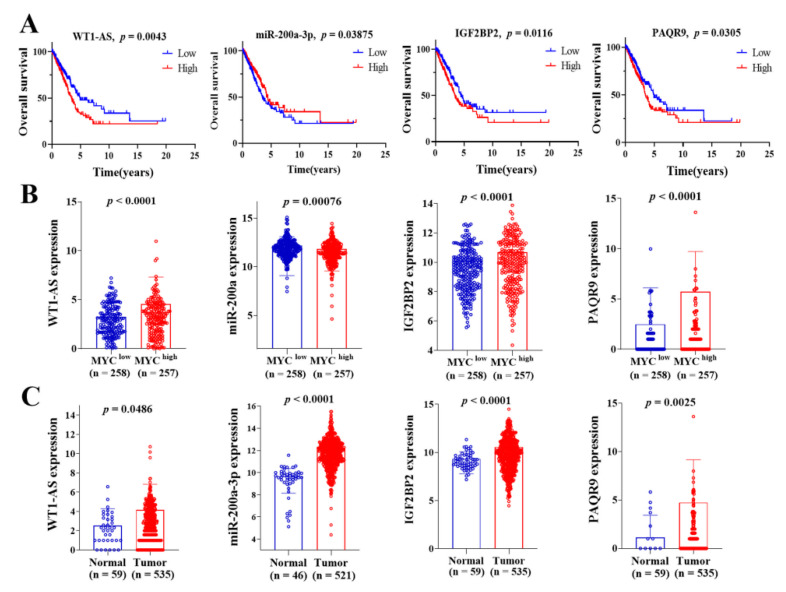
Correlation analysis of differentially expressed RNAs in the ceRNA network with overall survival of LUAD patients. (**A**) The high-expression and low-expression value of differentially expressed RNAs were compared by the Kaplan–Meier survival curve for the LUAD patient cohort; (**B**) The expression pattern of differentially expressed RNAs in LUAD samples with c-Myc^high^ and c-Myc^low^ groups; (**C**) The expression pattern of differentially expressed RNAs in LUAD samples and adjacent-normal lung tissues. The RNAs expression value was logarithmized with log2. *p* < 0.05 was deemed as statistically significant.

**Figure 4 cells-11-00025-f004:**
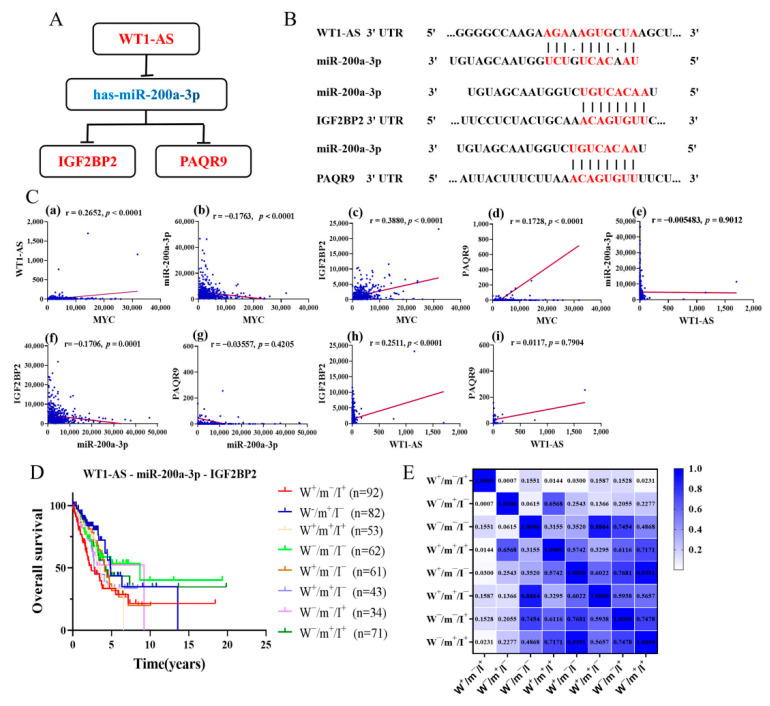
Correlation analysis and validation of WT1-AS, miR200a, and IGF2BP2 expression. (**A**) A schematic model of ceRNA. Blue indicates downregulation and red indicates upregulation. (**B**) Base pairing between miR-200a-3p and the target site in the WT1-AS, IGF2BP2, and PAQR9 predicted by Tarbase and TargetScan, respectively. (**C**) Correlation analysis between WT1-AS, miR200a, IGF2BP2, and PAQR9 in LUAD. (**D**,**E**) Correlation analysis of different expression patterns of WT1-AS–miR-200a-3p–IGF2BP2 with OS of LUAD patients and with a heat map of the significance of differences.

**Figure 5 cells-11-00025-f005:**
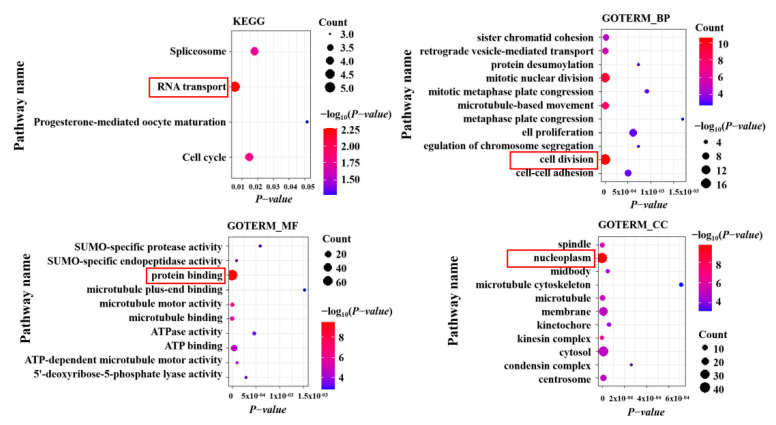
Functional enrichment analysis (including KEGG and GO) of IGF2BP2-associated genes in LUAD.

**Figure 6 cells-11-00025-f006:**
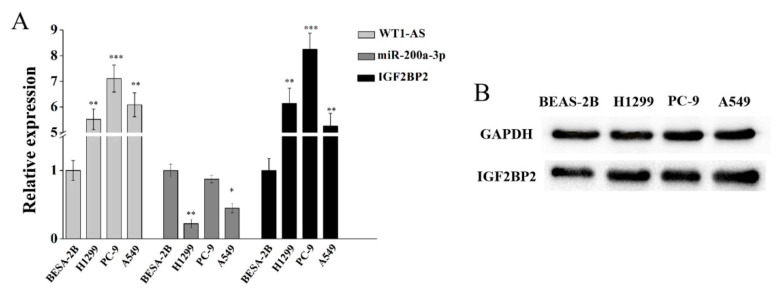
Validation of regulatory axis gene expression. (**A**) RT-PCR analysis detects the expression of WT1-AS, miR-200a-3p, and IGF2BP2 in LUAD cell lines. (**B**) The expression of IGF2BP2 protein in LUAD cell lines was detected by Western blot. * *p* < 0.05, ** *p* < 0.01, *** *p* < 0.001, by unpaired Student t test compared with control group.

## Data Availability

The data involved in this study were included in this manuscript and Appendix A.

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
