# Peer review of "WT1-AS/IGF2BP2 Axis Is a Potential Diagnostic and Prognostic Biomarker for Lung Adenocarcinoma According to ceRNA Network Comprehensive Analysis Combined with Experiments"

_cells, 2021, doi:10.3390/cells11010025_

Round 1

Reviewer 1 Report

The article by MingXi Jia and colleagues entitled “WT1-AS/IGF2BP2 Axis is a Potential Diagnostic and Prognostic Biomarker for Lung Adenocarcinoma According to ceRNA Net- work Comprehensive Analysis Combined with Experiments” tries to establish the WT1-AS/IGF2BP2 axis as a potential prognostic biomarker in lung adenocarcinoma. Overall, this study has some weakness but it could be used as starting point for a detailed analysis of this pathway. The authors have to improve the manuscript before it can be published in the journal “Cells”.

My major concerns are:

- The authors should not use abbreviations in the abstract.

- The authors have to introduce all abbreviations before using these. Of course, an abbreviation should be only introduced if it is used later in the text. After introducing an abbreviation, only the abbreviation should be used in the text.

- The citations 2-4 are too old and the authors have to prove their statements with up-to-date citations. Citation 1 not well suited as a prove and has to be replaced by a more relevant citation.

- Very often citations for statements are missing, e.g. lines 49, 52, 408 (of note this list is not complete).

- The authors state “….lncRNA competes directly or indirectly with miRNA for bind- 52 ing, which will eventually lead to a weakened interaction between miRNA and 53 mRNA”. This is true but the authors have to add the fact that lncRNAs are more often involved in DNA transcription and gene regulation (for details see e.g. DOI: 10.3389/fonc.2018.00226).

- In line 64 the authors must be more precise – from which tissues are the used cancer samples.

- It is neither nice nor necessary to read “Lu et al. ….” or “Miao et al. ….” etc. if somebody is interested in the name of the first author (s)he will find this information in the Reference list. The authors have to rephrase all these parts.

- The “material and methods” part is extremely poorly. The authors have to improve this part significantly. City and countries must be added to all companies, very often even the company is not mentioned for reagents and/or equipment. Another question is why the location of ThermoFisher differs. On top concentration for used solutions and reagents are missing.

- Primer sequences for GAPDH are missing.

- Figure 2B is too small and must be presented in a manner that the labelling is easy to read.

- The content from lines 300 and 301 should be better presented as a table.

Author Response

Dear professor:

I would like to thank you for your helpful review of my paper. Your comments are extremely useful, and I really appreciate your kindness. We have studied the comments carefully and made corrections which we hope meet with approval. The manuscript has modified according the comments.

Sincerely

Wen Li

Reviewer 2 Report

Authors have tried to identify MYC-related RNAs. They have identified WT1-AS/IGF2BP2 as a diagnostic marker for lung cancer.

However, the manuscript suffers from several problems.

1) How were TCGA data were normalized.

2) How many RNAs have MYC binding sites in their promoter.

3) Figs 1-4 already known, nothing novel.

4) Fig. 3 is not convincing.

5) Quality of research is very poor.

6) Fig. 7 doesn't add anything, it's all correlative.

Author Response

(The authors gave the same response as above.)

Reviewer 3 Report

First of all, I would like to thank the Authors for their effort in presenting the results of their study. The paper is interesting and original, but I would like to make a comment that I think might help to improve the work and better understand its overall objectives.

  1. Please make some efforts to improve the conclusion because the current version seems not elaborative. I suggest the Authors to add the section on limitations and future perspective.

Author Response

(The authors gave the same response as above.)

Round 2

Reviewer 1 Report

The authors have improved the manuscript significantly and the current version can be published.

Reviewer 2 Report

Authors have improved manuscript